# The influence of traceability label trust on consumers' traceability pork purchasing behavior: Based on the moderating effect of food safety identification

**Zengjin Liu** [1‡]*, **Zhuo Yu** [2‡], **Jing Zhao** [2], **Xibing Han** [3], **Caixia Li** [4], **Ning Geng** [3]*, **Meilian Yu** [1]

**1** Shanghai Academy of Agricultural Sciences, Shanghai, China, **2** School of Management, Ocean University of China, Qingdao, Shandong, China, **3** College of Public Administration, Shandong Normal University, Jinan, China, **4** Faculty of Environmental Science, Nagasaki University, Nagasaki, Japan

‡ ZL and ZY are co-first authors on this work.
* liuzengjin200632@126.com (ZL); gengning@sdnu.edu.cn (NG)

**Data Availability Statement:** All relevant data are within the manuscript and its Supporting Information files.

## Abstract

Based on 908 consumer questionnaire data from 15 urban areas in Shanghai, we use the binary logit model to empirically analyze the impact of traceability label trust on consumers' traceable pork purchase behavior and the moderating effect of food safety identification. After constructing the theoretical analysis framework, this paper verifies it from the two aspects of statistical analysis and econometric analysis and tests the robustness of the final results. The results show that: first, traceability label trust has a significant positive impact on consumers' traceability pork behavior. Second, food safety identification can significantly strengthen and promote this process. Third, consumers' purchasing behavior is significantly positively affected by traceable pork consumption scenarios and price labels, but the permanent elderly in the composition of family members significantly negative impact on it. Therefore, we put forward relevant policy suggestions, such as strengthening the knowledge popularization and publicity based on the advantageous commodity attributes of traceable pork, carrying out food safety knowledge popularization education, and enhancing consumers' risk perception and food safety identification ability.

## 1. Introduction

In recent years, the increase in population has put enormous pressure on food production and food safety [1], and food safety has been a global issue affecting public health with far-reaching political implications [2]. Since 2006, China has paid more attention to food safety during the 11th and 12th Five-Year Plan periods, issuing documents such as the Food Safety Law and the National Plan for Food Safety Supervision System to strengthen food safety protection [3, 4]. As the main meat product consumed by Chinese residents daily, whether the food quality and safety of pork can be effectively guaranteed is an important issue related to the national economy and people's livelihood. Food safety is a major public concern in China with far-reaching

**Funding:** This work was supported by the National Natural Science Foundation of China under the project "Study on the quality and safety effects of pork traceability system based on coupled incentive of regulation and reputation: theory and empirical evidence" (71603169). The funders had no role in study design, data collection and analysis, decision to publish, or preparation of the manuscript.

**Competing interests:** The authors have declared that no competing interests exist.

political implications [2]. Food quality and safety is also a worldwide problem, mainly caused by environmental pollution, enterprise production and operation, and government supervision failure [5] and the root cause is a market failure caused by information asymmetry [6, 7]. The longer the food supply chain, the higher the degree of information asymmetry [7, 8]. To eliminate information asymmetry and reduce food safety risks, governments and scholars have explored a variety of food safety risk management tools. Food traceability systems are a powerful solution to food safety issues [9, 10], and have become the main tool and prerequisite for ensuring meat quality and preventing food safety risks. Food traceability aims to reduce the risk of foodborne illness by facilitating recalls of food and feed products and providing consumers with targeted information [11]. Long-term unsafe food problems and short-term animal disease outbreaks can positively affect the willingness to consume traceable food [12]. Due to the negative impact of food scandals in China and the growing awareness of food safety, consumers and governments are constantly seeking healthier and safer food [13] Amid high-profile food scares, health concerns, and the threat of imperfect and asymmetric information, the Chinese pork industry faces growing consumer demand for assurance of quality and production methods in domestic and export markets [14].

The subsequent structure of this paper is as follows: Section 2 covers the Literature Review, Section 3 describes the Methodology, Section 4 presents the Survey and Data, Section 5 deals with the Statistical Analysis, Section 6 discusses the Model Estimation Results and Analysis, Section 7 concludes with the Conclusions.

## 2. Literature review

A review of the relevant literature reveals that food safety itself carries reputational attributes [15], and consumers are more inclined to purchase certified (with traceability) by reputational attributes, with more detailed traceability information [16–18], and more safe foods [19, 20]. Food labeling is in the interests of food producers and consumers, furthermore, it is seen as a policy tool to aid in the transition to a healthier and more sustainable food system [21]. The concepts of origin identification, certificate of conformity, traceability, and green organic certification involved in food safety information labels are important tools to eliminate information asymmetry in the food market. Since the mid-to-late 1990s, China has begun to explore the construction of a traceability system for pork and other foods, mainly involving the animal identification and disease traceability system of the Ministry of Agriculture and Rural Affairs, the national agricultural product quality and safety traceability management information platform, and five batches of support from the commercial department to establish a city-level meat and vegetable circulation traceability platform in 58 cities. The pork traceability system in China mainly includes two ways: One is the mode of ordinary pork used in the system in government encouragement, and the other is the mode of high-grade pork integrated production in the participation of live pig slaughtering or processing enterprises. Consumers, as the last link in the production chain, study what factors influence their behavioral patterns that will allow the meat industry to better meet consumer expectations, needs and wants [22]. Consumers' preference, willingness to pay, and purchase behavior for traceable products are research areas with certain practical significance, which are helpful for the promotion and construction of traceability systems and the realization of long-term goals of food safety management.

There are barriers to e-applications in the Indian agri-food supply chain that are highly driven and lowly dependent [23]. In addition, questionnaires from 30 sugar mills in Uttar Pradesh and Uttarakhand, India, showed that the problems identified by the e-application are developing positive relationships and providing them with suggested IT solutions that will

contribute to the overall development of farmers involved in sugarcane production [24]. There is also a study reviewing different individual and integrated approaches such as Multi-Criteria Decision Analysis (MCDA), multivariate, etc. to measure the problems at various stages of the supply chain of the olive oil industry in different countries [25]. Actually, Problems in the supply chain of other industries, such as the apparel industry export order fulfillment difficulties the most sensitive factor and in turn affects operating costs, changes in marketing strategies, changes in consumer purchasing patterns, and other factors, thus affecting profitability and employment cutoffs [26].

At present, the research in the field of the traceable system in the academic circle mainly focuses on the micro level, which is divided from the perspective of relevant stakeholders, involving consumers, enterprises and pig farmers [27, 28]. On the one hand, based on the traceable commodity attributes and the investigator's stated preferences, the research on the additional willingness to pay of consumers is generally carried out using the hypothetical value method, the BDM experimental auction, the real choice experiment and other methods to measure the willingness to pay [29, 30]. Furthermore, they use econometric model to analyze its influencing factors. Consumers are willing to pay a significant positive price premium for food traceability [31], and despite heterogeneity, all consumers have some positive willingness to pay for the labeling attributes of traceable pork [32]. Consumers are willing to pay a higher average premium for traceability with detailed information than just having information [16–18]. On the other hand, it is to study consumers' preference for traceable pork. It has been shown that there is preference heterogeneity among consumers for different attributes of food information, and the higher the education and income, the more likely they are to purchase food with safety information attributes [33]. Consumers' preference for traceable pork information attributes has obvious hierarchy and heterogeneity. Food safety has the largest premium of all attributes, and combined with risk perception results in a generally higher premium paid [34]. Consumers' preferences and willingness to pay for traceability information and quality certification are significantly affected by age, household monthly income and education level [35]. Furthermore, there is a study about consumers' purchase intention. The information quality, perceived reliability and product diagnostics of food traceability systems affect consumers' perceived value. Perceived value is positively correlated with purchase intention [36]. Consumers purchase Pork willingness and pork consumption were primarily influenced by attitudes and habit intensity, while social norms and perceptions played a secondary role [37]. Finally, around the establishment and promotion of the traceability system, there is research on the behavior of slaughtering enterprises, pig farmers and other actors [38–40].

It can be seen that the research results on traceable pork have been relatively fruitful in the research fields of consumer preference, purchase intention, willingness to pay, and the willingness and behavior of relevant actors of farmers and slaughterhouses. However, there are relatively few studies on consumers' traceable pork purchase behavior. Relevant studies mainly take pork purchasing behavior as an example, combined with the bait effect, it is believed that the bait effect of diet clean label is the strongest, followed by price, breeding mode and breeding time [41]; Both appearance and traceability produce stronger decoy effects than price, while traceability produces stronger decoy effects than appearance[42].

In terms of pork consumption, there is an inconsistency between consumers' purchase intention and actual purchase behavior [43]. The purchasing drivers of Chinese meat consumers include meat safety certification and meat perception of health [44]. This provides a good reference and reference for the study, but there is still room for further research. With the deepening of the construction of agricultural product traceability systems such as the pilot construction of the pork traceability system, the feasibility of empirical research on consumers' purchases of traceable pork has been continuously enhanced. Since consumers' self-reported

preferences are different from their real decisions [45], it may be more meaningful to distinguish the previous studies in hypothetical situations and based on the conclusions obtained in real situations. Therefore, starting from the moderating effect of food safety identification, this paper studies the influence of traceability label trust on consumers' traceable pork purchase behavior. This can enrich the theoretical understanding of this research field to a certain extent and has certain research value.

## 3. Methodology

### 3.1 Theoretical analysis

In order to analyze the influence of traceability label trust on consumers' traceable pork purchase behavior, the theoretical model of this paper is constructed based on the relevant theories of information source trust, income consumption theory, consumer behavior theory, etc., and related research results (Fig 1). Based on trust theory, this study argues that consumers' trust in traceability labeling plays a crucial role in their purchasing decisions [46]. When consumers believe that traceability systems are reliable and effective in ensuring food safety, they are more likely to purchase traceable pork products This study examines how food safety identification moderates the effect of trust in traceability labeling on purchasing behavior. The study also draws on consumer behavior theory, which emphasizes the role of perceptions, attitudes, and decision-making processes in influencing how consumers respond to marketing stimuli [47]. Social cognitive theory suggests that consumer behavior is influenced by their observations of others, their sense of self-efficacy, and their perception of the consequences of their actions [48]. In the context of purchasing traceable pork, researchers examined how consumers' perceptions of others' purchasing behaviors and self-efficacy related to food safety

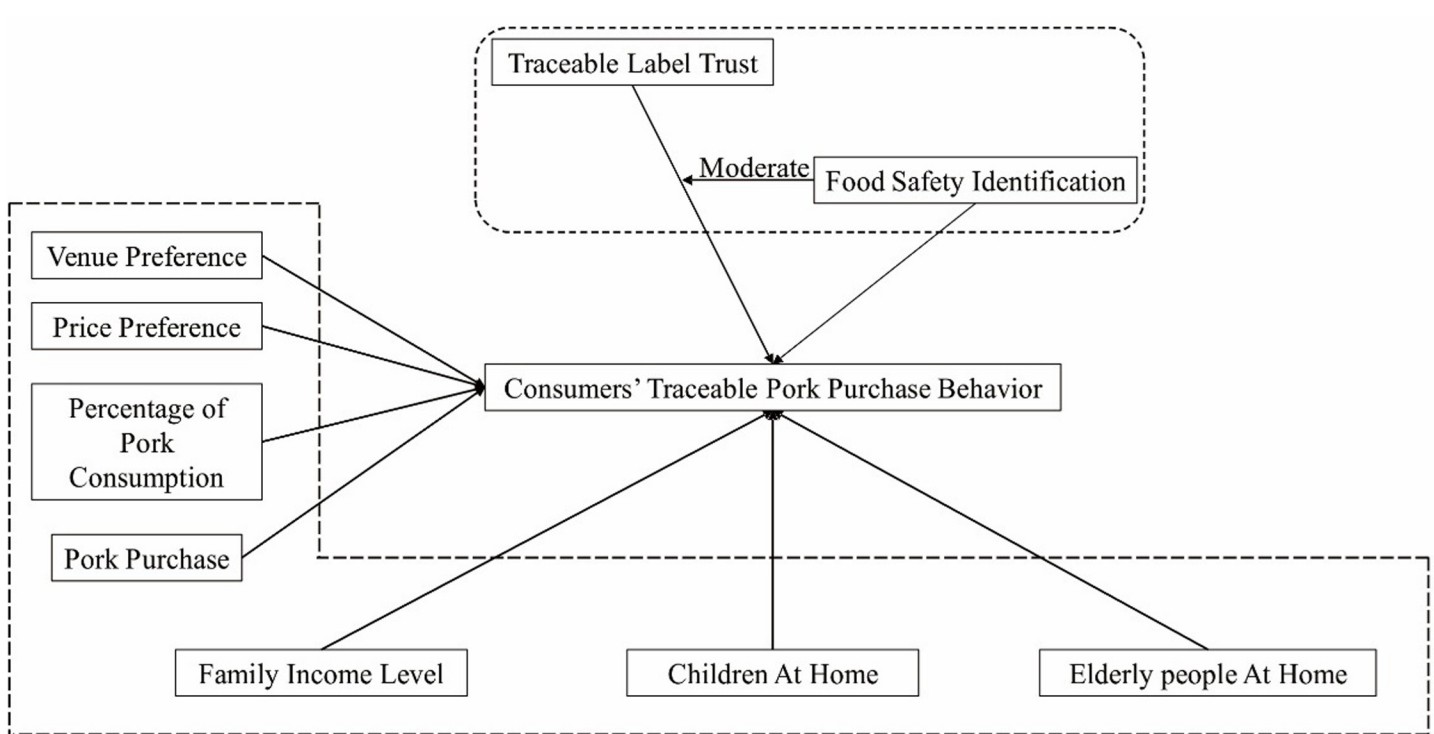

**Fig 1. Theoretical model of influencing factors of consumers' traceable pork purchasing behavior.** Source: Author Composition.

influenced their own purchasing decisions. This paper incorporates consumers' venue preference, price preference, household pork consumption proportion, pork purchase volume, income level, and the situation of children and the elderly in the family into the theoretical system that influences traceable pork purchase behavior. Further, we mainly analyze the influence of traceability label trust on this behavior, and try to interpret the role of food safety identification in this process.

In general, pork is not only involved in the field of personal consumption, but also a household consumer product to a large extent. Consumers' traceable pork purchase behavior is affected to a certain extent by consumers' personal preferences. When other factors remain unchanged, the correlation between consumption scenarios and corresponding commodities is strong. At this time, consumers' purchasing behavior is more susceptible to the influence of long-standing habits or cognition, and their preferences for attributes such as venue and price will have a certain impact on them. Of course, considering that family members go about their daily lives together and pork is a household consumer product, consumers play the role of buyers under the family division of labor. The factors involved in their decision-making are not limited to themselves, but are more affected by family needs or abilities' influences. Under the condition that other factors remain unchanged, the level of household income affects its consumption behavior and plays a fundamental role in the process of needing effectuation. Furthermore, as a special group in the family life cycle, the elderly or children's potential needs for the special attributes of some consumer goods have to be comprehensively considered. In addition to other influencing factors in the above theoretical system, consumers' traceable pork purchase behavior is mainly affected by the trust of traceability labels. Combined with theoretical analysis, the following research hypotheses are proposed.

**Research hypothesis 1:** Traceability label trust has a significant impact on consumers' traceable pork purchase behavior. The more trust in the traceability label, the greater the possibility of purchase behavior. For most countries, the rank order of food (pork) driving attributes that consumers care most about are health and nature, sensory quality, price, and animal welfare [49]. The role of food labels is to inform potential consumers about the quality of goods and to alleviate trust issues between consumers and other stakeholders [50]. Credit features play an important role in modern food marketing systems. Consumers' food choices are increasingly influenced by trust cues [15]. Traceability information is a solution for communicating safety practices, increasing consumer trust [51], and is used as a tool to help ensure food safety and quality and gain consumer confidence[52, 53]. By providing quality and safety information throughout the food process from farm to fork, traceability systems can reduce consumer uncertainty stemming from the food purchasing process, enhancing price premiums and facilitating purchasing behavior [54]. As an information source, traceable labels have two characteristics trust and attractiveness. In today's context of a serious lack of social integrity, high trust is an attraction in itself. As an information source, traceable labels have two characteristics trust and attractiveness. In today's context of a serious lack of social integrity, high trust is an attraction in itself. The food traceability system can restore consumer confidence by improving consumer trust, thereby increasing the possibility of consumers buying traceable food.

**Research hypothesis 2:** Food safety identification can significantly affect the effect of traceability label trust on consumers' traceable pork purchase behavior. **Research Hypothesis 2–1:** Food safety recognition plays a positive regulatory effect in this process. **Research hypothesis 2–2:** Food safety identification has a significant impact on consumers' traceable purchase behavior. Despite the limited ability to identify safe food, Chinese consumers generally have positive attitudes towards it, especially in terms of safety, quality, nutrition and taste, and they are willing to pay more for safe food [55]. At the consumer level, food safety identification is

based on risk perception and the ability to identify food safety issues. In the actual purchase process, it is embodied in the final manifestation of the current consumer concern and judgment for the food safety situation. Food incident information leads to a decrease in trust, and negative information about food safety increases perceived risk to common foods and hazards, and indirectly increases overall perception of food safety risk [56], and food safety concerns also contribute to its Generating more positive buying attitudes. Therefore, in the process that traceable labels are used as a medium to transmit product quality information to influence consumers' decision-making on traceable pork purchase behavior, consumers' attention to the current product safety status may change their stated preferences for traceable label attributes and the actual demand, which there will be a certain adjustment effect.

### 3.2 The empirical model

Since the behavior of consumers to purchase traceable pork includes two choices, "purchased" and "not purchased", this is a typical categorical variable and is suitable for the Logit model. Therefore, the binary logit model is selected and the ML estimation method is used. In order to comprehensively and systematically study the influencing factors of consumers' traceable pork purchase behavior, and make the influence of traceability label trust on purchase behavior more credible. This paper will consider the influence of other explanatory variables and control variables such as income level and age, and combine the data characteristics to control the regional and time effects.

$$Ln \frac{p(Y = 1)}{1 - P(Y = 1)} = \alpha_0 + \beta_i(X\,Z\,M\,C\,O\,I\,K) + e \tag{1}$$

In formula (1), the explained variable Y represents the traceable pork purchase behavior of consumers, 1 represents the purchase, 0 represents the non-purchase, and β represents the parameter to be estimated of the model variable. As the core explanatory variable, X represents consumer traceability label trust, and M is introduced into the model as a moderating variable in the form of M and XM. In terms of pork consumption characteristics of consumers, variable Z includes four aspects: the proportion of pork consumption, the amount of pork purchased, price preference, and purchase place preference. In terms of family characteristics, variables C and O represent the number of resident elderly (elders over 60 years old) and resident children (under 15 years old) in the family, respectively, and I represents family income. K represents the control variables in the model, including the respondents' gender, age, education, and household registration. e represents the residual term, which, according to the model assumptions, obeys an asymptotically normal distribution. The meaning, assignment and descriptive statistical characteristics of model variables are shown in Table 1. The estimation of each model involved in this paper is realized by the software stata16.

## 4. Survey and data

With the continuous advancement of the construction of traceable pork pilot cities, the number of survivable samples of consumers in relevant areas involving traceable pork purchases is increasing, which also provides a certain basis for this paper. The data in this paper comes from the research team's research on Bao shan, Feng xian, Hong kou, Huang pu, Jia ding, Jin shan, Jing'an, Min hang, Pu dong, Pu tuo, Qing pu, Song jiang, Xu hui in Shanghai during May-June 2017 and July-October 2020. As the economic center of China, Shanghai has a large pork consumption market and high consumer demand for pork quality and safety. Moreover, Shanghai's pork market has a strong industrial scale and influence, and its pig slaughtering, processing enterprises and sales links are relatively mature and representative. Therefore, the

**Table 1. Variable selection, basic meaning and descriptive statistics.**

| variable type | variable name | Meaning and valuation | Mean value |
|---|---|---|---|
| Explained variable | | | |
| | Traceable pork purchases | Whether the respondent has purchased pork with information traceability label: Purchased = 1, never purchased = 0 | 0.463 |
| core explanatory variables | | | |
| | Traceability Label Trust | Strongly believe = 5, somewhat believe = 4, generally believe = 3, do not quite believe = 2, very disbelieve = 1 | 3.840 |
| Moderating variables | | | |
| | Food Safety Identification | When purchasing pork, the degree of concern about the quality and safety of pork: very concerned = 5, relatively concerned = 4, generally concerned = 3, not very concerned = 2, very not concerned = 1 | 3.933 |
| other explanatory variables | | | |
| | Percentage of pork consumption | The proportion of the respondents' current pork consumption in household livestock and poultry meat consumption: less than 10% = 1, 10–29% = 2, 30–49% = 3, 50–69% = 4, 70% and above = 5 | 2.686 |
| | Pork purchase quantity | The average monthly purchase amount of pork in the respondents' current households: 0–8 catties = 1, 9–12 catties = 2, 13–16 catties = 3, 17–20 catties = 4, over 20 catties = 5 | 1.840 |
| | Price preference | The main unit price of pork purchased by the respondents' households: | |
| | | 20 yuan/catties or less = 1 | 0.337 |
| | | 30 yuan/catties or more = 1 | 0.279 |
| | venue preference | Respondents mainly buy pork places: | |
| | | Supermarket = 1 | 0.632 |
| | | Specialty store, convenience store, small supermarket = 1 | 0.305 |
| | | farmers market = 1 | 0.490 |
| | | Network platform = 1 | 0.120 |
| | Children at home | Number of children (under 15 years old) living together in the family | 0.371 |
| | Elderly people at home | The number of elderly people (60 years old and above, referring to the elders) in the family who live together | 0.265 |
| | family income level | Annual household income in the previous year (after tax): Below 100,000 = 1, 100,000–490,000 = 2, above 500,000 = 3 | 1.972 |
| control variable | | | |
| | Gender | Respondent gender: male = 1, female = 0 | 0.498 |
| | Age | Respondent's actual age | 36.132 |
| | Household | registration Local = 1, non-local = 0 | 0.512 |
| | Education | Junior high school and below = 1, technical secondary school/high school = 2, junior college = 3, undergraduate = 4, postgraduate and above = 5 | 3.476 |

Source: Author Composition

study of the construction of Shanghai pork traceability system can help to understand the safeguard measures of pork quality and safety in regions with higher economic development level, and can also provide reference for other cities and regions. A questionnaire survey was conducted by consumers in 15 urban areas of, Yang Pu and Chang Ning. After strict screening such as logical verification, 908 valid questionnaires were finally obtained. There were 240 valid questionnaires for 2017 and 668 valid questionnaires for 2020. In order to ensure the usability of the questionnaire data, this research was conducted in the form of face-to-face interviews in the field. The investigators were professors and students from the Shanghai Academy of Agricultural Sciences and the School of Economics and Management of Shanghai Ocean University. Moreover, we conducted special training and pre-survey of the

questionnaire for the investigators before the formal survey. The respondents were all permanent residents of Shanghai and consumers who had purchased pork. Before the formal questionnaire survey, the respondents were asked whether they had purchased pork, and the questionnaire survey was conducted only when they received an affirmative answer. Respondents who completed the questionnaire were given a towel gift with a market price of about 10 yuan to mobilize the respondents to participate in the survey. At the same time, this also helps to ensure the quality of questionnaire completion. The survey locations were mainly chosen in supermarkets, farmers' markets and their neighborhoods. We considered the sample size of the questionnaire survey in each urban area according to the distribution of permanent residents in each urban area.

## 4.1 Sample basic characteristics

The basic characteristics of the research samples in this paper are shown in Table 2. From the perspective of gender distribution, the ratio of males and females is relatively balanced, indicating that the gender division of labor in purchasing pork and other foods in the households of the survey respondents is not obvious. In terms of age distribution, the proportion of young and middle-aged people under the age of 30 and 30–39 years old among the survey respondents is relatively large, accounting for 39.65% and 29.3% respectively. In terms of educational background, more than 60% of the survey respondents have undergraduate and postgraduate degrees, and the remaining educational levels account for a relatively balanced proportion of about 10%. This phenomenon may be related to the siphoning effect of megacities on high-quality labor.

**Table 2. Basic characteristics of the samples.**

| variable | Options | Ratio (%) | variable | Options | Ratio (%) |
|---|---|---|---|---|---|
| gender | male | 49.78 | household registration | local | 51.21 |
| | female | 50.22 | | nonlocal | 48.79 |
| age | less than 30 years old | 39.65 | Family resident elders (60 years old and above): | Zero person | 73.57 |
| | 30 to 39 years old | 29.30 | | one person | 26.32 |
| | 40 to 49 years old | 12.33 | | Two person | 0.01 |
| | 50 to 59 years old | 7.71 | Children in the family (under 15 years old): | Zero person | 62.89 |
| | 60 years old and above | 11.01 | | One person | 37.11 |
| education | Junior high school and below | 11.34 | Household Income in the Previous Year (After Taxes) | Below 100,000 yuan | 26.65 |
| | Secondary School/High School | 12.67 | | 10000 to 490000 yuan | 49.45 |
| | college | 15.86 | | More than 500,000 yuan | 23.90 |
| | Undergraduate | 37.33 | Average monthly household purchases of pork | 0 to 8 catties | 54.30 |
| | Graduate and above | 22.80 | | 9 to 12 catties | 22.47 |
| Proportion of pork consumption in household livestock and poultry meat consumption | Below 10% | 17.18 | | 13 to 16 catties | 13.22 |
| | 10% to 29% | 30.95 | | 17 to 20 catties | 4.96 |
| | 30 to 49% | 26.76 | | More than 20 catties | 5.07 |
| | 50 to 69% | 16.30 | unit price of pork often purchased | Below 20 yuan | 33.70 |
| | 70% and above | 8.81 | | 20 to 30 yuan | 38.44 |
| | | | | more than 30 yuan | 27.86 |

Source: Author Composition

The distribution of household registration can also effectively confirm this phenomenon, and the proportion of local and non-local permanent residents is relatively balanced. In terms of the composition of the family members of the respondents, the elders and children accounted for 26.33% and 37.11% respectively, and the respondents' family population was relatively small. In terms of household income distribution, the household income of the respondents in the previous year was 100,000–490,000 and 500,000 or more, accounting for 49.45% and 23.90%, respectively, which is more in line with the income status of residents in the surveyed cities. From the perspective of consumer household pork consumption characteristics, the proportion of pork consumption in household livestock and poultry meat consumption accounted for 30–49% of the respondents accounted for 26.76%, and the proportion of consumption accounted for 10–29% accounted for 30.95% %, the proportion of consumption is 70% and above, which is 8.81%. In terms of the unit price of pork that is often purchased, 38.44% of the respondents spend 20–30 yuan, and 30% of the respondents are in less than 20-yuan and more than 30-yuan level, indicating that the actual payment level of pork by the respondents is relatively high.

## 5. Statistical analysis

### 5.1 Traceable label trust, food safety identification, consumer traceability of pork purchases

Through the questionnaire, do you believe that "the quality and safety of agricultural products with information traceability labels are more guaranteed than those without information traceability codes"? This item allows consumers to answer whether they trust the traceability labels based on their purchasing experience or subjective judgment. Through the question "Do you pay attention to the quality and safety of pork when you usually buy pork?" to measure consumers' food safety identification. Investigators defined traceable pork purchases based on consumers' subjective responses by carrying examples of commonly used information traceability label. In the sample, 46.25% of the respondents purchased traceable pork, 92.73% of the trust information traceability label, and 91.19% of consumers are more concerned about the quality and safety of pork. Under the premise of purchasing traceable pork, the proportion of trusting the traceability information code and paying attention to food safety is 90.95%, which is 81.90% higher than other cases. Comparing and analyzing the samples in 2017 and 2020, the proportion of respondents who trust the information traceability code and the proportion concerned about pork quality and safety have fluctuated and increased, while the proportion purchasing traceable pork has increased from 38.75% to 48.95%.

### 5.2 The relationship between traceability label trust and consumers' traceable pork purchase behavior

In order to have a more intuitive understanding of the relationship between traceability label trust and consumers' traceable pork purchase behavior, a cross-analysis of the two was conducted, and the results are shown in Table 3. Under the premise of not trusting traceability labels, consumers who did not purchase traceable pork accounted for 81.82%. Under the premise that consumers purchase traceable pork, the proportion of consumers who trust the traceable label is about 30% higher than the proportion that they do not trust. Therefore, there is a certain correlation between traceable label trust and consumers' traceable pork purchase behavior. We undertake a chi-square test and also test the Cramers V. The results show a very significant difference between traceability label trust and consumers' traceable pork purchase behavior. Consumers who trust the label are more likely to have traceable pork purchase

**Table 3. Cross-analysis of traceability label trust and consumer traceability pork purchase behavior.**

| Traceability label | Traceable pork purchase | | No traceable pork was purchased | |
|---|---|---|---|---|
| | Frequency | Proportion (%) | Frequency | Proportion (%) |
| trust | 408 | 48.46 | 434 | 51.54 |
| distrust | 12 | 18.18 | 54 | 81.82 |
| Total | 420 | 46.25 | 488 | 53.75 |
| Chi-square and Cramers V test | | | | |
| | Value | df | P | |
| Pearson | 22.564 | 1 | 0.000 | |
| Cramers V | 0.158 | - | 0.000 | |

Source: Author Composition

behavior. The Cramers V is used to determine whether there was a significant association between traceability label trust and consumers' traceable pork purchase behavior. [57].

## 5.3 The relationship between food safety identification and traceability label trust

In order to have a more intuitive understanding of the relationship between food safety identification and follow-label trust, a cross-analysis of the two was conducted, and the results are shown in Table 4. Among the survey respondents, the number of samples who trust the traceability label and pay attention to pork quality and safety is far greater than that of those who do not trust the label and do not pay attention to pork quality and safety. When consumers are concerned about pork quality and safety, their trust in traceability labels is 87.2% higher than their distrust. Therefore, there is a certain correlation between concerns about pork quality and safety and trust in traceability labels. The results of chi-square test and Cramers V test both indicate a significant difference between food safety identification and follow-label trust. Combined with the previous analysis, traceability label trust, food safety identification and traceable pork purchase behavior show a certain positive correlation, but whether there is a clear causal relationship still needs to be tested by an econometric model.

**Table 4. Cross analysis of food safety identification and traceability label trust.**

| Food Safety Identification | Trust traceability label | | Distrust traceability label | |
|---|---|---|---|---|
| | Frequency | Proportion (%) | Frequency | Proportion (%) |
| Concerned | 775 | 93.60 | 53 | 6.40 |
| Not concerned | 67 | 83.75 | 13 | 16.25 |
| Total | 842 | 92.73 | 66 | 7.27 |
| Chi-square and Cramers V test | | | | |
| | Value | df | P | |
| Pearson | 10.499 | 1 | 0.001 | |
| Cramers V | 0.108 | - | 0.001 | |

Source: Author Composition

# 6. Model estimation results and analysis

## 6.1 Model selection, estimation results and analysis

Although the previous article has carried out statistical analysis on the relationship between traceability label trust and consumers' traceable pork purchase behavior, it has not comprehensively considered or controlled for the influence of other factors. Therefore, it is not scientifically reasonable to explain that traceability label trust has a significant impact on consumers' traceable pork purchase behavior, and it is also necessary to quantitatively analyze the impact of traceability label trust on consumers' traceable pork purchase behavior.

This article uses the Shanghai consumer survey data in 2017 and 2020, and fails to track the individual respondents. There are certain differences in individual characteristics in the data, which belong to pseudo-panel data. Therefore, it is necessary to test and verify the model setting. Model (1) is a panel logit model with random effects, and model (2) is a mixed panel logit model. According to the LR test results of model (1) in Table 5, the null hypothesis of rho = 0 should be accepted. Because the panel-level variance component is unimportant when rho is zero, the panel estimator can be considered indistinguishable from the mixed cross-section estimator. At this time, it is no longer necessary to perform the Hausman test to compare the fixed effect and the random effect, and the mixed Logit model should be selected as the main model.

**Table 5. Estimated results of model form selection.**

| Variable name | Model name | |
|---|---|---|
| | **Model (1)** | **Model (2)** |
| Traceability label trust | 0.561*** (0.1091) | 0.508*** (0.0864) |
| Food Safety identification | 0.178** (0.0878) | 0.159** (0.0781) |
| Percentage of pork consumption | 0.073 (0.0723) | 0.065 (0.0651) |
| Pork purchase quantity | -0.005 (0.0735) | -0.003 (0.0667) |
| Price preference | | |
| Below 20 yuan/catties | -0.555*** (0.1935) | -0.499*** (0.1687) |
| 30 yuan/catties or more | 0.139 (0.1917) | 0.129 (0.1687) |
| Venue preference | | |
| Hypermarket | 0.474*** (0.1741) | 0.437** (0.1543) |
| Specialty stores, convenience Stores, small supermarkets | 0.045 (0.1759) | 0..0338 (0.1596) |
| Agricultural market | 0.062 (0.1664) | 0.056 (0.1508) |
| Network platform | 0.496* (0.2568) | 0..431* (0.2251) |
| Children at home | 0.121 (0.1709) | 0.104 (0.1541) |
| Elderly people at home | -0.312* (0.1871) | -0.287* (0.1683) |
| Family income level | -0.002 (0.1146) | 0.010 (0.1036) |
| Control variable | | |
| Age | 0.001 (0.0067) | 0.001 (0.0061) |
| Gender | 0.262* (0.1594) | 0.239* (0.1436) |
| Education | -0.047 (0.0710) | -0.044 (0.0645) |
| Register | 0.201 (0.1732) | 0.1807 (0.1562) |
| LR test Pseudo R2 or Wald test | LR test of rho = 0: chibar2(01) = 1.17 Wald chi2(17) = 46.89*** | LR chi2(17) = 93.06*** Pseudo R$^2$ = 0.0742 |

Source: Author Composition

Note: The numbers outside the brackets are the estimated coefficients, and the numbers in the brackets are the standard errors; ***, ** and * indicate significance at the 1%, 5% and 10% levels, respectively.

## 6.2 Main effect and significant variable impact analysis

After adopting the mixed logit model, considering the integrity of the model construction, the influence of variables that do not change with individual changes cannot be ignored. Therefore, the time effect and the regional effect are introduced respectively, and model (3) and model (4) are constructed, and the estimated results are shown in Table 6. Judging from the estimation results of model (2), model (3) and model (4), the overall goodness of fit and significance of the model are better. Combined with the significance level of variables, the analysis is as follows. Consistent with the conclusion of the previous statistical analysis, traceability label trust has a significant positive impact on consumers' traceable pork purchase behavior, which further validates the previous theoretical analysis. After pork is traceable, whether it is pre-quality assurance or post-responsibility traceability, the stakeholders in each link of the industry chain can be quickly identified, which is convenient for consumers to protect their own rights and interests, and builds consumers' trust in traceability labels, which further stimulates the possibility of traceability. Trace pork purchases.

In terms of venue preference and price preference, large supermarkets and online platforms have a significant positive impact on consumers' traceable pork purchase behavior, while purchasing pork with a relatively low relative price level has a significant negative impact. This

**Table 6. Estimated results of the model considering temporal effect and regional effect.**

| Variable name | Model name | |
|---|---|---|
| | **Model (3)** | **Model (4)** |
| Traceability label trust | 0.506*** (0.0865) | 0.554*** (0.0897) |
| Food Safety identification | 0.170** (0.0784) | 0.173** (0.0805) |
| Percentage of pork consumption | 0.083 (0.0660) | 0.073 (0.0677) |
| Pork purchase quantity | -0.001 (0.0668) | -0.001 (0.0685) |
| Price preference | | |
| Below 20 yuan/catties | -0.364** (0.1814) | -0.309* (0.1863) |
| 30 yuan/catties or more | 0.089 (0.1747) | 0.162 (0.1799) |
| Venue preference | | |
| Hypermarket | 0.469*** (0.1555) | 0.469*** (0.1593) |
| Specialty stores, convenience Stores, small supermarkets | 0.029 (0.1599) | -0.03667 (0.1649) |
| Agricultural market | 0.054 (0.1513) | 0.004 (0.1556) |
| Network platform | 0.377* (0.2266) | 0.449* (0.2341) |
| Children at home | 0.126 (0.1552) | 0.140 (0.1620) |
| Elderly people at home | -0.341** (0.1716) | -0.314 * (0.1753) |
| Family income level | 0.013 (0.1037) | 0.024 (0.1064) |
| Control variable | | |
| Age | 0.003 (0.0062) | 0.004 (0.0064) |
| Gender | 0.210 (0.1446) | 0.249* (0.1487) |
| Education | -0.067 (0.0656) | -0.079 (0.0686) |
| Register | 0.178 (0.1565) | 0.274* (0.1635) |
| Time effect | —— | —— |
| Regional effect | | —— |
| LR test and pseudo R2 | LR chi2(18) = 97.14*** Pseudo R$^2$ = 0.0775 | LR chi2(32) = 125.80*** Pseudo R$^2$ = 0.1006 |

Source: Author Composition

Note: The numbers outside the brackets are the estimated coefficients, and the numbers in the brackets are the standard errors; ***, ** and * indicate significance at the 1%, 5% and 10% levels, respectively.

may be related to the current model of Chinese enterprises participating in the construction of the traceability system. Enterprises expect to create brand differences through traceability, and then achieve product premium or high-priced sales. At the same time, some consumers' understanding of traceable pork mainly comes from corporate propaganda, which makes the current scene of traceable pork purchases more in large supermarkets and online platforms. Consumers in consumption scenarios such as convenience stores, small supermarkets, and farmers' markets are more inclined to purchase pork with low-end price levels to a certain extent, and are more sensitive to the additional price paid for ordinary pork participating in the traceability system.

In terms of family members, the elderly who live in the family have a significant negative impact on consumers' traceable pork purchase behavior. We believe that a possible explanation for this phenomenon is that in the division of labor among urban families, the resident elderly will help to buy daily food and take care of housework, so as to reduce the burden of life and work for their children. Affected by living habits such as diligence and thrift, they will pay more attention to whether the price is affordable when purchasing pork and other meat, so they are relatively less likely to purchase traceable pork.

### 6.3 Analysis of moderating effects based on food safety identification

According to the estimation results of model (2), model (3), and model (4), it can be seen that food safety identification as a moderating variable has a significant impact on consumers' traceable pork purchase behavior. Now the influence between food safety identification and traceability label trust is introduced, which further demonstrates the direction and extent of its influence on traceability label trust on consumers' traceable purchase behavior. As shown in Table 7, model (5), model (6), and model (7) under the premise of considering time and regional effects, the moderating effect of food safety identification is very significant and credible. Specifically, food safety identification plays a strengthening and promoting moderating role in the process of traceability label trust positively affecting consumers' traceable pork purchase behavior. To a certain extent, this proves the theoretical analysis of the previous article.

### 6.4 Robustness test of empirical results

To ensure the robustness of the model estimation results, robustness checks are required. The purpose of a robustness test is to ensure that the results of the empirical analysis (sign, significance) do not change with changing parameter settings. Generally, the estimation is re-estimated by changing the measurement method of the core variables, changing the model setting, etc. If the results of the empirical analysis remain unchanged, it is considered to have passed the robustness test. Therefore, on the one hand, we changed the measurement method of food safety identification variables, and selected consumers' confidence in the quality and safety of the currently purchased pork to measure (Food Safety Identification 2), and used the logit model to re-estimate. On the other hand, combined with the original variables, use the OLS mixed regression or probit model to re-estimate the model. The results of the robustness test in this paper are shown in Table 8. Combined with the changes in the signs and significance levels of key explanatory variables, it can be seen that the above empirical analysis results on the influence of traceability label trust on consumers' traceable pork and the moderating effect of food safety identification are relatively robust.

## 7. Conclusions

Combined with a questionnaire survey on consumers' traceable pork purchase behavior in Shanghai, this study constructs a theoretical framework based on the influence of traceability

**Table 7. Estimated results of the moderating effect model of food safety recognition.**

| Variable name | Model name | | |
|---|---|---|---|
| | Model (5) | Model (6) | Model (7) |
| Traceable label trust* Food safety identification | 0.615** (0.2406) | 0.621** (0.2413) | 0.662*** (0.2463) |
| Food Safety identification | 0.151* (0.0860) | 0.161* (0.0863) | 0.170* (0.0883) |
| Percentage of pork consumption | 0.053 (0.0642) | 0.071 (0.0650) | 0.059 (0.0666) |
| Pork purchase quantity | 0.011 (0.0658) | 0.016 (0.0660) | 0.014 (0.0674) |
| Price preference | | | |
| Below 20 yuan/catties | -0.488*** (0.1659) | -0.348* (0.1781) | -0.303* (0.1821) |
| 30 yuan/catties or more | 0.119 (0.1712) | 0.081 (0.1724) | 0.152 (0.1771) |
| Venue preference | | | |
| Hypermarket | 0.458*** (0.1518) | 0.491*** (0.1530) | 0.494*** (0.1564) |
| Specialty stores, convenience Stores, small supermarkets | 0.015 (0.1571) | 0.011 (0.1573) | -0.053 (0.1621) |
| Agricultural market | 0.060 (0.1485) | 0.053 (0.1489) | 0.010 (0.1530) |
| Network platform | 0.484** (0.2220) | 0.427* (0.2236) | 0.497** (0.2307) |
| Children at home | 0.065 (0.1514) | 0.089 (0.1526) | 0.123 (0.1587) |
| Elderly people at home | -0.305* (0.1659) | -0.364** (0.1691) | -0.337* 0.1723 |
| Family income level | -0.016 (0.1018) | -0.013 (0.1019) | -0.009 (0.1043) |
| Control variable | | | |
| Age | -0.0001 (0.0060) | 0.002 (0.0061) | 0.003 (0.0063) |
| Gender | 0.186 (0.1407) | 0.156* (0.1418) | 0.189 (0.1454) |
| Education | -0.041 (0.0633) | -0.066 (0.0645) | -0.072 (0.0671) |
| Register | 0.272* (0.1534) | 0.269* (0.1537) | 0.369** (0.1604) |
| Time effect | | —— | —— |
| Regional effect | | | —— |
| LR test and pseudo R2 | LRchi2(17) = 62.41*** Pseudo R² = 0.0498 | LR chi2(18) = 67.03*** Pseudo R² = 0.0535 | LR chi2(32) = 91.72*** Pseudo R² = 0.0733 |

Source: Author Composition

Note: The numbers outside the brackets are the estimated coefficients, and the numbers in the brackets are the standard errors; ***, ** and * indicate significance at the 1%, 5% and 10% levels, respectively.

label trust on consumers' traceable pork purchase behavior under the moderating effect of food safety identification. Through statistical analysis, quantitative analysis, etc. Empirical research, draws the following conclusions with certain robustness. First, there is a certain correlation between traceability label trust and traceable pork purchase behavior. With the increase in consumers' trust in traceability labels, the proportion of consumers who purchase traceable pork will increase. Similar results were found by Liang et al. [58] Second, traceability label trust has a significant positive impact on consumers' traceable pork purchase behavior. Food safety identification can significantly strengthen, promote and other regulating effects on this influence process. The specific performance is that when consumers buy pork, the more attention they pay to the quality and safety of pork, the more they can strengthen the functions of traceability labels by providing information reference and determining after-the-fact responsibilities, building trust and further stimulating potential consumers to buy process of behavior. After changing the measurement method of the variable of food safety identification, that is, consumers are more assured of the quality and safety of the pork currently purchased,

**Table 8. Robustness test of the empirical results.**

| Variable name | Change estimation method | | | | Substitution variable | |
|---|---|---|---|---|---|---|
| | Probit | Ols | Probit | Ols | Logit | Logit |
| Traceable label trust | 0.309*** (0.0519) | 0.112*** (0.0185) | | | 0.488*** (0.0865) | |
| Food safety identification 1 | 0.095** (0.0474) | 0.035** (0.0174) | 0.091* (0.0530) | 0.035** (0.020) | | |
| Food safety identification 2 | | | | | 0.292*** (0.0896) | 0.276*** (0.0981) |
| Traceable label trust* Food safety identification 1 | | | 0.366** (0.1464) | 0.133** (0.0542) | | |
| Traceable label trust* Food safety identification 2 | | | | | | 0.684*** (0.2541) |

Source: Author Composition

Note: The numbers outside the brackets are the estimated coefficients, and the numbers in the brackets are the standard errors; ***, ** and * indicate significance at the 1%, 5% and 10% levels, respectively.

which also strengthens the positive impact of traceability label trust on consumers' traceable pork purchase behavior. This is in line with the findings of Matzembacher' s research that food traceability systems can increase the level of consumer utility and further enhance consumer trust in food quality [15]. Third, the current consumers' purchase behavior of traceable pork is significantly positively affected by consumption scenario preferences and price preferences. Consumers are more inclined to buy traceable pork in supermarkets or online platforms, and when the price level of household pork is lower, it is significantly less likely to purchase traceable pork. In addition, the resident elderly in the composition of family members will have a significant negative impact on it, which is also largely in line with the findings of existing studies on traceable foods [59].

## Policy recommendations

Based on the above research conclusions, the starting point is to accelerate the construction of the pork traceability system and improve the marketization level of traceable pork, and put forward the following policy implications: (1) Give full play to the role of traceability label trust, combined with the experience of traceable pork city pilot work, strengthen the popularization and publicity work based on the traceable pork advantage commodity attributes. (2) Relying on community and grass-roots service organizations, carry out food safety knowledge popularization education, enhance consumers' risk perception and food safety identification capabilities, give full play to the regulatory role of food safety identification, and lay a foundation for expanding the traceable pork market space. (3) Appropriately increase government investment, continue to promote the construction of pork traceability system and quality certification, and focus on promoting the information richness, coverage area, publicity and promotion of ordinary pork in agricultural product traceability information management, so as to make traceability system reduce food quality and safety risks can benefit all sectors of society. (4) Relying on the advantages of enterprises under the traceable pork system, the demonstration drives the whole industry to realize the large-scale and standardized breeding of live pigs, steadily expand the scale of traceable pork supply, effectively control and improve the quality and safety of pork, and gradually realize the comprehensive and effective traceability of pork in my country.

## Limitations and future work

In this study, the impact of traceability labeling trust on consumers' traceable pork purchasing behavior was verified through a questionnaire survey of 908 consumers in Shanghai. A Logit

model was applied to analyze the impact of traceability labeling trust on consumers' traceable pork purchasing behavior. Through existing studies, more studies have focused on the study of consumers' willingness to pay for traceable pork, and fewer studies have examined the impact of traceability labeling trust on consumers' traceable pork purchasing behavior, and this study also focuses on the time effect and regional effect, as well as the moderating effect of food safety identification, which is the innovation of this study. This study also focuses on the time and regional effects of this influence and the moderating effect of food safety identification, which is the innovation of this study. However, we also recognize that at this particular historical stage of the construction of China's pork traceability system, constrained by the fact that the level of the construction of China's pork traceability system is not yet high, in addition to investigating the effect of traceability label trust on consumers' traceable pork purchasing behaviors, we would like to validate the effect of traceability label querying behaviors and the content of the information of traceability label querying on consumers' continuous purchasing behaviors of traceable pork, which is also our next investigation and research plan.

## Supporting information

**S1 File.**
(XLSX)

## Author Contributions

**Conceptualization:** Meilian Yu.

**Data curation:** Zengjin Liu, Zhuo Yu, Jing Zhao.

**Formal analysis:** Zengjin Liu, Zhuo Yu, Jing Zhao.

**Funding acquisition:** Meilian Yu.

**Supervision:** Zengjin Liu, Ning Geng.

**Validation:** Zengjin Liu, Ning Geng.

**Writing – original draft:** Zengjin Liu, Zhuo Yu, Jing Zhao.

**Writing – review & editing:** Xibing Han, Caixia Li, Meilian Yu.

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
