## [Decision Letter · Decision Letter 0]

29 Nov 2023

PONE-D-23-35868The Influence of Traceability Label Trust on Consumers' Traceability Pork Purchasing Behavior: Based on the Moderating Effect of Food Safety IdentificationPLOS ONE

Dear Dr. Liu,

Thank you for submitting your manuscript to PLOS ONE. After careful consideration, we feel that it has merit but does not fully meet PLOS ONE’s publication criteria as it currently stands. Therefore, we invite you to submit a revised version of the manuscript that addresses the points raised during the review process.

We look forward to receiving your revised manuscript.

Kind regards,

Joshua Orungo Onono, Ph.D

Academic Editor

PLOS ONE

 [This work was supported by the National Natural Science Foundation of China under the project “Study on the quality and safety effects of pork traceability system based on coupled incentive of regulation and reputation: theory and empirical evidence” (71603169)].  

5. PLOS requires an ORCID iD for the corresponding author in Editorial Manager on papers submitted after December 6th, 2016. Please ensure that you have an ORCID iD and that it is validated in Editorial Manager. To do this, go to ‘Update my Information’ (in the upper left-hand corner of the main menu), and click on the Fetch/Validate link next to the ORCID field. This will take you to the ORCID site and allow you to create a new iD or authenticate a pre-existing iD in Editorial Manager. Please see the following video for instructions on linking an ORCID iD to your Editorial Manager account: " ext-link-type="uri" xlink:type="simple">https://www.youtube.com/watch?v=_xcclfuvtxQ".

6. We note that Figure 2 in your submission contain [map/satellite] images which may be copyrighted. All PLOS content is published under the Creative Commons Attribution License (CC BY 4.0), which means that the manuscript, images, and Supporting Information files will be freely available online, and any third party is permitted to access, download, copy, distribute, and use these materials in any way, even commercially, with proper attribution. For these reasons, we cannot publish previously copyrighted maps or satellite images created using proprietary data, such as Google software (Google Maps, Street View, and Earth). For more information, see our copyright guidelines: http://journals.plos.org/plosone/s/licenses-and-copyright.

Additional Editor Comments:

Thank you for submitting the paper for consideration for publications, reviewers have recommended that you do major revision before it can be reconsidered for publication.

Reviewers' comments:

Reviewer's Responses to Questions

**Comments to the Author**

1. Is the manuscript technically sound, and do the data support the conclusions?

Reviewer #1: Partly

Reviewer #2: Partly

2. Has the statistical analysis been performed appropriately and rigorously? 

Reviewer #1: Yes

Reviewer #2: No

3. Have the authors made all data underlying the findings in their manuscript fully available?

Reviewer #1: Yes

Reviewer #2: No

4. Is the manuscript presented in an intelligible fashion and written in standard English?

Reviewer #1: Yes

Reviewer #2: Yes

5. Review Comments to the Author

Reviewer #1: Author have done a good job but only essential corrections needed before publications. I will be happy to revise it and check all these.

1.Introduction: There should be section wise representation of all headings included in a single para at the end.

2.LR: I unable to see the research gaps and objectives. Please include latest references which will give you an idea of work in different sectors. Few important papers needs to be added: DOI: 10.4018/978-1-4666-5202-6.ch022 ; https://doi.org/10.1504/IJBPSCM.2015.073770 ; Green supply chain management: A case of sugar industry in India. In: Proceedings of national conference on emerging challenges for sustainable business ; https://doi.org/10.3390/w15081498 ; https://doi.org/10.1504/IJLSM.2020.103862 ; https://doi.org/10.1016/j.jclepro.2022.133311 ; https://doi.org/10.1504/IJVCM.2017.086838 ; DOI: 10.4018/IJISSCM.287133 ; https://doi.org/10.1007/s10479-021-04371-y ; https://doi.org/10.1007/s12063-022-00270-y ; https://doi.org/10.1177/09722629221107238 ; https://doi.org/10.1108/IJPPM-02-2022-0080 ; https://doi.org/10.1504/IJLEG.2023.130236 ; https://doi.org/10.1504/IJMP.2018.090829 ; https://doi.org/10.1007/s10668-022-02456-7 ; Supply chain performance of sugar industry using Regression analysis. International Journal of Engineering Technology Management and Applied Sciences.

3.Methodology: Please check for proper citation of figures and tables in the text. In addition, there is no source given for figure and tables. If it is self then author should cite as “Source: Author Composition” or similar. Few of the latest papers should be cited in the text like: https://doi.org/10.1007/s10479-023-05431-1 ; https://doi.org/10.1111/opec.12287 ; https://doi.org/10.3390/horticulturae8111018 ; https://doi.org/10.1504/IJVCM.2022.122164 ; https://doi.org/10.1177/097226292211432 ; https://doi.org/10.3390/w15122274

4.Results and Discussion should be seperate heading and must showcase the outputs and compare with other studies also.

5.Implications should be added in this (Political, Practical and Social if any).

6.Conclusion part is written in very short. Compare your study with other authors and write in conclusion and extend a bit more.

7.Limitations and future work should be included as a seperate heading or may be a part of conclusion also in the last para.

8.Please make a proper structure of your manuscript like aligning with Introduction, LR, Gaps, Objectives, Methods and Materials, Analysis, Results and Discussion, Implications, Conclusion, Limitations and Future Scope and References.

Reviewer #2: Comments

Introduction

1.On page 3, lines 71-77, you are to give citations to the various studies listed.

2. From lines 117-140, the theoretical basis for this study is not strong and based mainly on comments and hypotheses by the authors and not literature. The authors should answer the question of what is the theory grounding this study. What theory is expected to be validated or otherwise at this study's end and how it fits well into its objectives?

3.The conceptual framework for the figure 1 should be relooked into. First, the food safety identification is to play a moderating role between traceability lable trust and purchase behaviour. However what is presented does not depict that. Again, the other variables included in the framework have not been explained in the text and hence need to be explained.

Methodology

1.Explain why the logit model and not the probit model were used in the methodology. What makes Logit fit the probit model better for this study? Given the moderating effect, why didn’t you use Structural Equation modelling but Logit?

2.Under the study area, the authors should justify why the study area is the best-fit place for this study compared to other potential areas in China.

3.The sampling procedure here too is not clear. Was it convenient sampling? If yes, how did you get the people who have had pork purchase experience vis a vis ow many did you come across that had not pork purchase experience? Be clear on the sampling of respondents and the interviews.

Results and Discussions

1.Table 2 is clumsy, and following through with the figures is challenging. The authors should redo it for clarity.

2.Lines 270-272's argument is far-fetched. You just did cross-tabulation. For you to make that statement, undertake a chi-square test of the two crosstabulated variables and also test the Cramers V to know the level of relationship in percentage terms.

3.The same should be done for Table 4.

4.The relevance of Table 6 and its difference from Table 5 is not explained. The authors should do well to explain.

5.The authors are unclear; the differences among models 1-5 and the moderating effect that was analysed are unclear from the results of Table 7. I suggest structural equation modelling is used to moderate the effect to a more precise picture since the data is not a complete panel data.

Conclusions and Recommendations

1.Conclusions and recommendations could be better after the suggested corrections have been done.

6. PLOS authors have the option to publish the peer review history of their article (what does this mean?). If published, this will include your full peer review and any attached files.

Reviewer #1: No

Reviewer #2: No

---

## [Author Response · Author response to Decision Letter 0]

27 Feb 2024

Thanks for reviewers and editor comments. We have responded all of your advice one by one. Thanks again!

---

## [Decision Letter · Decision Letter 1]

10 Jun 2024

The Influence of Traceability Label Trust on Consumers' Traceability Pork Purchasing Behavior: Based on the Moderating Effect of Food Safety Identification

PONE-D-23-35868R1

Dear Dr.,

We’re pleased to inform you that your manuscript has been judged scientifically suitable for publication and will be formally accepted for publication once it meets all outstanding technical requirements.

Kind regards,

Sandro Vieira Soares, Ph.D.

Academic Editor

PLOS ONE

Comments from PLOS Editorial Office:

We note that one or more reviewers has recommended that you cite specific previously published works. As always, we recommend that you please review and evaluate the requested works to determine whether they are relevant and should be cited. It is not a requirement to cite these works. We appreciate your attention to this request.

Reviewers' comments:

Reviewer's Responses to Questions

**Comments to the Author**

Reviewer #3: All comments have been addressed

Reviewer #4: All comments have been addressed

2. Is the manuscript technically sound, and do the data support the conclusions?

Reviewer #3: Yes

Reviewer #4: Yes

3. Has the statistical analysis been performed appropriately and rigorously? 

Reviewer #3: Yes

Reviewer #4: Yes

4. Have the authors made all data underlying the findings in their manuscript fully available?

Reviewer #3: Yes

Reviewer #4: Yes

5. Is the manuscript presented in an intelligible fashion and written in standard English?

Reviewer #3: Yes

Reviewer #4: Yes

6. Review Comments to the Author

Reviewer #3: (No Response)

Reviewer #4: This study investigates the impact of trust in traceability labels on the purchasing behavior of traceable pork in Shanghai, utilizing survey data and binary logistic analysis. It is observed that trust in labels has a significant positive impact on purchasing behavior, with this effect being moderated by food safety identification. The methodological analyses are solid, employing robust statistical models for hypothesis validation. The article is well-structured, including a comprehensive literature review and a detailed discussion of the results. The manuscript is recommended for acceptance, as the quality of the writing is suitable for a high-impact publication. This work significantly contributes to the literature on purchasing behavior and food safety.

7. PLOS authors have the option to publish the peer review history of their article (what does this mean?). If published, this will include your full peer review and any attached files.

Reviewer #3: No

Reviewer #4: No

---

## [Editor Report · Acceptance letter]

21 Jun 2024

PONE-D-23-35868R1 

PLOS ONE

Dear Dr. Liu, 

I'm pleased to inform you that your manuscript has been deemed suitable for publication in PLOS ONE. Congratulations! Your manuscript is now being handed over to our production team.

Kind regards, 

on behalf of

Dr. Sandro Vieira Soares 

Academic Editor

PLOS ONE